# Investigation on Flaw Evolution of Additively Manufactured Al_2_O_3_ Ceramic by In Situ X-ray Computed Tomography

**DOI:** 10.3390/ma15072547

**Published:** 2022-03-30

**Authors:** Naijia Cai, Qiaoyu Meng, Keqiang Zhang, Luchao Geng, Rujie He, Zhaoliang Qu

**Affiliations:** 1Institute of Advanced Structure Technology, Beijing Institute of Technology, Beijing 100081, China; 3120190002@bit.edu.cn (N.C.); m_qiaoyu@163.com (Q.M.); zhangkeq@126.com (K.Z.); 2College of Engineering, Peking University, Beijing 100871, China; genglc@pku.edu.cn

**Keywords:** stereolithographic additive manufacturing, Al_2_O_3_ ceramic, flaw evolution, in situ X-ray computed tomography

## Abstract

The additive manufacturing process may create flaws inside ceramic materials. The flaws have a significant influence on the macroscopic mechanical behavior of ceramic materials. In order to reveal the influence of flaws on the mechanical behavior of additively manufactured ceramic, flaw evolution under mechanical loads was studied by in situ X-ray computed tomography (XCT) in this work. In situ compression XCT tests were conducted on stereolithographic additively manufactured Al_2_O_3_ ceramic. The three-dimensional full-field morphologies at different compressive loads were obtained. The evolution of flaws, including pores, transverse cracks, and vertical cracks, during compressive loading was observed. The number and volume of pores, transverse cracks, and vertical cracks were extracted. It was found that most pores and transverse cracks tend to be compacted. However, high compressive loads cause vertical cracks near the upper surface to expand, leading to the failure of the specimen. Real flaws with morphological and positional information were introduced into the finite element models. The influence of different types of flaws on the mechanical behavior is discussed. It was found that vertical cracks have a greater influence on mechanical behavior than do transverse cracks under compression. The presence of transverse cracks contributes to the evolution of vertical cracks. This study may be helpful for process optimization and performance enhancement of additively manufactured ceramic materials.

## 1. Introduction

Due to their excellent combined properties of high strength, high hardness, and good chemical inertness, advanced ceramics have received considerable interest and attention for structural applications [1]. The high hardness and inherent brittleness pose difficulties in machining, which limits the design and application of complex-shaped ceramic components [2]. Additive manufacturing (AM), as a near-net shape processing approach, produces three-dimensional objects by adding material layer by layer. It provides an effective way to enable the fabrication of complex-shaped components without expensive machining [3,4]. AM has been widely adopted in the field of ceramic manufacturing, and various AM technologies for ceramics have been developed [5,6]. Among these techniques, stereolithography additive manufacturing technology has the advantage of high precision, excellent surface quality, and fast printing speed, and it has successfully been applied for fabricating complex-shaped ceramics, such as ZrO_2_, Al_2_O_3_, SiC, etc. [7,8,9].

Stereolithography additive manufacturing technology for ceramics generally comprises shaping, debinding (binder burnout), and sintering processes [7,8,9]. During these processes, flaws in ceramics such as pores, cracks, etc., may appear [10,11]. For example, insufficient binder burnout and/or a non-optimal sintering process may lead to large pores [10]. Defects may also be induced during the detachment of the green body from the building platform [11]. It is widely accepted that ceramics are brittle and sensitive to flaws. The presence of flaws may decrease the strength value and increase the strength dispersion [12,13]. It is necessary to study the influence of manufacturing flaws on the mechanical properties of additively manufactured ceramics, which is crucial for the optimization of these mechanical properties. Thus, mechanical characterization based on a flaw study is an important task and has been the focus of several studies [11,14,15]. Schlacher et al. [14] reported that the influence of process-typical defects, such as delaminations or large pores, on the uniaxial strength of AM Al_2_O_3_ depends on the relation between the printing and loading directions. The microstructural characteristics and fracture strength of yttria-stabilized tetragonal zirconia polycrystal fabricated by stereolithographic additive manufacturing and traditional subtractive manufacturing were studied. Although both materials showed similar microstructures considering grain size and phase composition, significant differences in critical defects were observed [15]. Another study by Harrer et al. concluded that the strength of yttria-stabilized zirconia ceramic obtained via the lithography-based ceramic manufacturing process was limited by flaws that were introduced by processing and handling [11]. The influence of flaws on mechanical properties was studied by fractographic analysis based on scanning electron microscopy (SEM) in the above studies. However, pre-mechanical loading may change the state of flaws or even generate new flaws. The initial state and evolution of manufacturing flaws should also be obtained before and during mechanical loading, respectively.

SEM, as a commonly used method for microstructural characterization, provides only two-dimensional information of outer surfaces. To obtain three-dimensional information of the whole body, successive sectioning and subsequent microscopy are adopted. However, the sectioning process may cause damage to materials. X-ray computed tomography (XCT) technology can obtain internal information of materials and has become an established technique for nondestructive analysis [16,17,18]. In our previous study, the effects of solid loading on the defects and mechanical performance of additively manufactured zirconia ceramic were studied by XCT analysis [19]. Saâdaoui et al. successfully applied XCT analysis for the non-destructive detection of typical defects in additive-manufactured zirconia. A good agreement between XCT data, bending strength measurements, and fractographic analysis demonstrated the suitability of XCT for both defect detection and predictive mechanical strength estimation [20]. In the above studies, systematic XCT analysis was carried out before mechanical tests. The aim of this work was therefore to adopt XCT analysis to investigate the flaw evolution of AM ceramic during the mechanical tests.

In this paper, compression tests with in situ XCT were conducted on additively manufactured Al_2_O_3_ ceramics. The flaw evolution behaviors under compressive loads were investigated. The influence of different types of flaws on the mechanical behavior is discussed. It is believed that this study may provide guidance for the process optimization and performance enhancement of additively manufactured ceramic materials.

## 2. Experimental Section

### 2.1. Material and Specimens

A schematic for the preparation of additively manufactured Al_2_O_3_ ceramics is presented in Figure 1. The photosensitive slurry used to fabricate the Al_2_O_3_ green bodies was reported in a previous work [8]. Al_2_O_3_ coarse grain particles of 50 vol.% (3.8 g/cm^3^, d_50_ = 10.34 μm, Zhongzhou Alloy Material Co., Ltd., Shanghai, China) were used as the raw material to prepare Al_2_O_3_ ceramic slurry. Al_2_O_3_ fine grain particles (3.8 g/cm^3^, d_50_ = 1.05 μm, Zhongzhou Alloy Material Co., Ltd., China) were added as fine grains. Al_2_O_3_ slurry was dispersed in a pre-photosensitive resin suspension that consisted of photosensitive resin monomers (HDDA and TMPTA), photoinitiator (1 wt% TPO), dispersant (2 wt% KOS110), and sintering additives (3 wt% TiO_2_ and 1 wt% MgO). The photosensitive slurry was mixed via ball milling at 400 rpm for 24 h. Al_2_O_3_ green bodies were then fabricated using stereolithographic additive manufacturing equipment with a 405 nm ultraviolet light projector (AutoCera, Beijing 10dim Tech., Co., Ltd., Beijing, China). The light intensity, exposure time, and slicing thickness were set to 14,000 μw/cm^2^, 4 s, and 100 μm, respectively. After that, the Al_2_O_3_ green bodies were debound at 550 °C for 2 h (a heating rate of 0.5 °C/min) and pre-sintered at 1000 °C for 2 h in a furnace (FMJ-07/11, HeFei Facerom Thermal Co., Ltd., Hefei, China). Finally, the debound bodies were sintered at 1650 °C for 3 h in air in a furnace (FMJ-05/17, HeFei Facerom Thermal Co., Ltd., Hefei, China). Based on the above process, Φ 5 mm × 12.5 mm (diameter × height) sintered specimens that were built with layers in the height direction were prepared for in situ XCT tests. The microstructures of the specimens were observed using a scanning electron microscope (JEOL JSM-7500F, JEOL Ltd., Tokyo, Japan).

### 2.2. In Situ XCT Experiments

In situ compression tests were conducted on additively manufactured Al_2_O_3_ ceramics using an in situ X-ray computed tomography apparatus assembled in our laboratory, as shown in Figure 2 [21]. This apparatus has a 300 W microfocus X-ray source producing X-rays with tube voltages from 30 to 225 kV. The specimen was rotated in a cone beam of X-rays, and the transmitted radiographic projections were imaged via a flat panel detector. A material testing machine with two synchronous rotating motors was specially designed for mechanical loading and 360° rotation of the specimen with the aim to obtain projection images at different angles. To shorten the distance between the X-ray source and the specimen and achieve high resolution, customized grips for the round table shape were adopted. The specimen was first scanned at the reference (unloaded) state, then loaded and scanned under compression. The specimen was loaded using displacement control at a quasi-static rate. The compression tests were regularly interrupted to perform tomographic scans. In situ XCT scans were collected at different compressive loads. To determine the compressive loads corresponding to scans, compression tests on five specimens were conducted. The specimens all failed at compressions ranging from 10,000 to 15,000 N. Thus, the selected compressive loads included 3000 N, 6000 N, and 10,000 N. After reaching the selected load, the applied displacement was fixed. For each scan, the specimen was rotated by 360° in steps of 0.36°, giving 1000 exposures. The exposure time was 3 s per projection, with each scan requiring 50 min. The effective voxel size was 8 μm, and the tube voltage was 140 kV. Reconstructed images were obtained by a filtered back-projection algorithm. Three-dimensional visualization, segmentation, and quantification were performed using the image processing software Avizo.

## 3. Experimental Results and Discussion

Figure 3 shows SEM images of the microstructures of the stereolithographic additively manufactured Al_2_O_3_. The relative density of the sintered specimen was 96.63 ± 0.52%. The layer-by-layer characteristic was observed on the surface of the material in layer-by-layer processing of stereolithography additive manufacturing. It can be seen that the material had a relatively fine-grained microstructure. The grain size in the specimen ranged from 1 to 6 μm. The crystallization of Al_2_O_3_ particles was also observed.

Load and displacement data were recorded during the compression tests. A typical load–displacement curve is shown in Figure 4a. Hollow circles indicate scan points at different compressive loads. The scan points are numbered sequentially as the load increases. It was found that some relaxation occurred during the constant displacement (holding) periods applied to the specimens during each scan. Two-dimensional XCT images at different compressive loads were obtained, as shown in Figure 4b. Manufacturing flaws, including pores and cracks, were observed in the additively manufactured Al_2_O_3_ specimens.

Three-dimensional volume rendering of total flaws at different compressive loads was obtained by stacking two-dimensional images, as shown in Figure 5a. Stress concentration occurs when the cured layer is released from the membrane. Moreover, over-curing may also cause high stress due to the high input energy. Defective structures in the form of pores and microcracks emerge owing to the stress release during debinding and sintering. There are many manufacturing flaws inside additively manufactured Al_2_O_3_ ceramics, and the flaws evolve with increasing compressive load. At 10,000 N, flaws near the upper surface evolve into large flaws, resulting in the failure of the specimen. To quantitatively analyze the evolution behavior of the flaws during compressive loading, the number and volume of the total flaws at different compressive loads were also obtained, as shown in Figure 5b,c, respectively. It was found that the number of total flaws decreased with increasing compressive load. This may be due to the gradual compaction of some flaws as the load increases. The volume of total flaws firstly decreased in the range from 0 to 3000 N, slowly increased in the range from 3000 to 6000 N, and sharply increased in the range from 6000 to 10,000 N. This evolution behavior is complex and difficult to understand.

To gain insight into the evolution behavior of flaws with the load, the flaws were classified and analyzed. As mentioned before, flaws mainly include pores and cracks. The sphericities of pores were calculated to distinguish between pores and cracks. Flaws with a sphericity value less than 2.9 were classified as pores, and those with a sphericity value larger than 2.9 were classified as cracks. Three-dimensional volume renderings of the pores at different compressive loads were individually extracted and are shown in Figure 6a. It was found that the holes were mainly located in the central area inside the specimen. As the compressive load increased, pore features were increasingly less frequently observed. Furthermore, the number and volume of the pores at different compressive loads were also counted and calculated, as shown in Figure 6b,c, respectively. It is clear from Figure 6b,c that the number and volume of the pores both decreased with increasing compressive load. Specifically, they both decreased sharply in the range from 6000 to 10,000 N. The number and volume of pores at 10,000 N were significantly reduced compared to those at other loads. This is mainly due to the compaction effect of the pores.

As is widely known, the relationship between crack orientation and loading direction may have an important influence on the crack evolution. Thus, cracks were further subdivided according to crack orientation. A coordinate system was created and is shown in Figure 4a. The *z*-axis was located along the height direction of the specimen and parallel to the loading direction. Cracks with a maximum size greater than 200 μm in the *z*-direction were defined as vertical cracks, and those with a maximum size less than 200 μm in the z-direction were defined as transverse cracks. Figure 7 shows three-dimensional volume renderings and the number and volume of vertical cracks at different compressive loads. From Figure 7a, it can be seen that there were vertical cracks near the upper surface of the specimen. There was almost no change in the morphology of vertical cracks from 0 to 6000 N. However, compressive loads generated tensile stresses at the tips of the vertical cracks and caused vertical cracks near the upper surface to expand. Large vertical cracks were observed at 10,000 N. Meanwhile, specimen failure occurred. The number and volume of the vertical cracks both changed very little from 0 to 6000 N, as shown in Figure 7b,c. From 6000 to 10,000 N, the number of vertical cracks sharply decreased, and their volume sharply increased. The decrease in number may be due to the compaction effect. It is worth noting that the volume of vertical cracks at 10,000 N was much larger than those at other loads, resulting from the formation of large cracks.

Three-dimensional volume renderings and the number and volume of transverse cracks at different compressive loads were also obtained, as shown in Figure 8. Some transverse cracks can be identified as delaminations, which are common in additively manufactured ceramics. Figure 8a shows the change in the morphology of transverse cracks during compression tests. Transverse cracks were mainly located in the peripheral area inside the specimen. It is clear that transverse cracks almost disappeared at 10,000 N. Figure 8b,c gives the number and volume of transverse cracks at different compressive loads. The number of transverse cracks decreased with increasing compressive load. As mentioned earlier, this may also have resulted from the compaction effect. Different from the change in number, the volume of the transverse cracks decreased from 0 to 3000 N, increased from 3000 to 6000 N, and decreased from 6000 to 10,000 N. The volume at 10,000 N was much smaller than those at other loads.

To explain the change in volume with the load, some transverse cracks with a large volume were chosen and extracted separately for analysis, as shown in Figure 9. Four transverse cracks were selected and labelled, as shown in Figure 9a. The volumes of each transverse crack were calculated at different compressive loads, as shown in Figure 9b. It was found that the change in volume was similar to that of all transverse cracks. The volumes of the representative transverse cracks also did not decrease monotonically with increasing load. The presence of flaws may be responsible for the non-monotonic change in volume. It may change the stress distribution inside the specimen and create a complex stress state rather than a simple compressive stress state.

Under compressive loads, most pores and transverse cracks tend to be compacted. However, the vertical cracks have an important influence on the failure of additively manufactured Al_2_O_3_ ceramic specimens. Tensile stresses on the tip of vertical cracks induced by compressive loads can lead to specimen failure.

## 4. Finite Element Analysis

Finite element analysis was adopted to further investigate the influence of different types of flaws on the mechanical behavior of the additively manufactured Al_2_O_3_ ceramic specimens. Four three-dimensional (3D) finite element models, as shown in Figure 10, were constructed to simulate the mechanical responses using ABAQUS commercial software. Model 1 was the ideal model without any flaws. The two vertical cracks near the upper surface were extracted and placed in Model 2. The transverse crack with the largest volume was extracted and placed in Model 3. Model 4 contained two vertical cracks and one transverse crack. The constitutive behavior of the specimen was perfectly elastic. A Drucker–Prager model and a shear damage model were chosen to define the damage and to simulate the fracture process of the material. The finite element models were meshed with four-node linear tetrahedron elements. A convergent solution with respect to the number of elements was confirmed. All the displacement and rotational degrees of freedom for the bottom surface of the specimen were constrained. A series of downward displacements were imposed on the upper surface of the specimen to simulate the loading process.

Although the displacement data could be obtained from the sensor of the material testing machine, the measured displacement was the displacement of the crossbeam rather than that of the specimen. Two-dimensional XCT images were used to calculate the true displacement of the specimens, as shown in Figure 11a. Representative features in the images were selected as marker points. The distances between the marker points at different compressive loads were calculated by counting the number of pixels. According to the geometric positions of the marker points, the heights of the specimen at different compressive loads were extracted. Then, the heights of adjacent loading steps were superimposed to obtain the true displacements at different compressive loads. The obtained true displacements were used to plot the load–displacement curve, as shown in Figure 11b. Moreover, the load–displacement curve based on the sensor data was also plotted in Figure 11b. It was found that the true displacements obtained from the images were much less than those obtained from the sensor. The true displacements obtained from the images were imposed on the upper surface of the specimens to simulate the loading process.

Load–displacement curves corresponding to the four models were obtained from the finite element analysis, as shown in Figure 12. The load–displacement curve obtained from the experiment was also plotted in Figure 12. It is clear that the load–displacement curve obtained from Model 4 including two vertical cracks and one transverse crack is the closest to that obtained from the experiment. The difference between the ideal model and this experiment is the biggest. The influence of vertical cracks on the load–displacement curve is greater than that of transverse cracks. In the presence of vertical cracks, the influence of transverse cracks on the load–displacement curve is reduced. The damage variable distribution for the models was also extracted and plotted in Figure 13. Crack propagation behavior could thus be simulated. Cracks extended outward along the edge of the contour, resulting from the tension and shear stresses at the edge of the crack. Figure 13a gives the damage variable distribution for Model 4 including two vertical cracks and one transverse crack. The propagation of vertical cracks near the upper surface was observed. This result is similar to the experimental result. By comparing Figure 13a and Figure 13b, we can conclude that the presence of transverse cracks contributes to the evolution of vertical cracks. It is possible that the transverse cracks weaken the stiffness of the specimen and increase the complexity of the stress.

## 5. Conclusions

In this paper, we investigated the influence of manufacturing flaws on the macroscopic mechanical behavior of additively manufactured Al_2_O_3_ ceramic. We conclude the following:(1)Stereolithography additive manufacturing technology was used to prepare Al_2_O_3_ ceramic specimens. In situ XCT compression tests were conducted on additively manufactured Al_2_O_3_ ceramic specimens. Details on the three-dimensional full-field morphology of the specimens at 0, 3000, 6000, and 10,000 N were obtained by stacking two-dimensional XCT images.(2)Different types of flaws, including pores, transverse cracks, and vertical cracks, were studied. The number and volume of the pores, transverse cracks, and vertical cracks were analyzed. It was found that the number of flaws tended to decrease with increasing compressive load, resulting from the compaction effect. At 10,000 N, vertical cracks near the upper surface expanded and formed large vertical cracks, leading to the failure of the specimen.(3)Real flaws with morphological and positional information were introduced into the finite element models to study the influence of different types of flaws on the mechanical behavior. The true displacements were obtained by analyzing XCT images. The failure mode obtained from the model including two vertical cracks and one transverse crack was similar to the experimental result. It was found that the influence of vertical cracks on the mechanical behavior was greater than that of transverse cracks. The presence of transverse cracks contributes to the evolution of vertical cracks.

## Figures and Tables

**Figure 1 materials-15-02547-f001:**
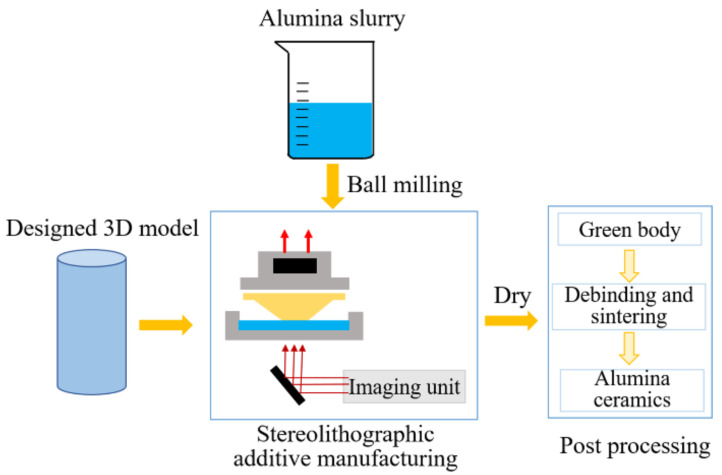
Schematic for the preparation of additively manufactured Al_2_O_3_ ceramics.

**Figure 2 materials-15-02547-f002:**
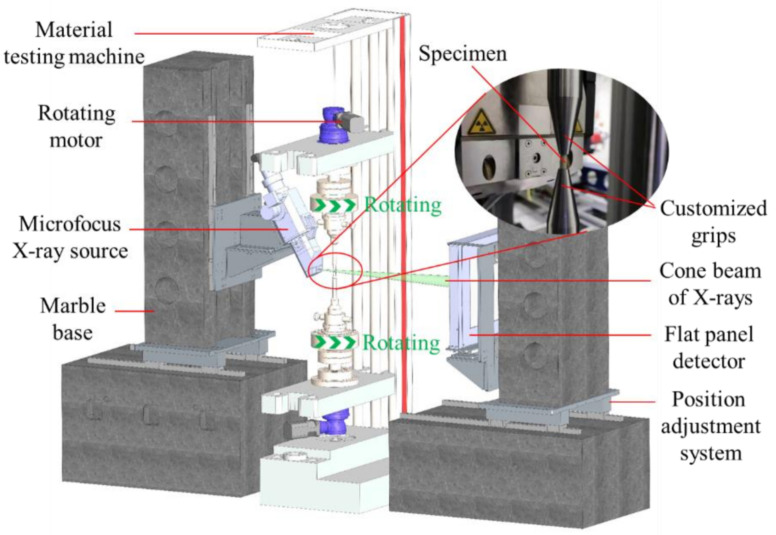
Schematic of the in situ X-ray computed tomography apparatus.

**Figure 3 materials-15-02547-f003:**
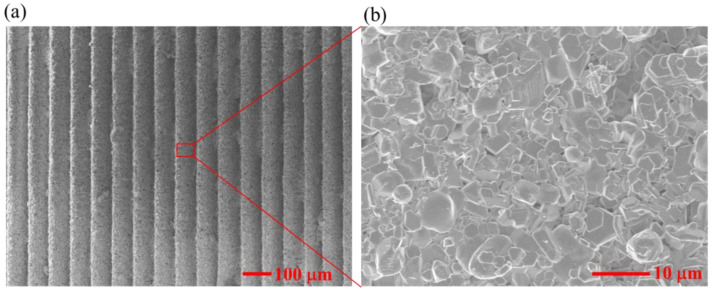
SEM images of the microstructures of stereolithographic additively manufactured Al_2_O_3_. (**a**) At the 100 μm scale; (**b**) At the 10 μm scale.

**Figure 4 materials-15-02547-f004:**
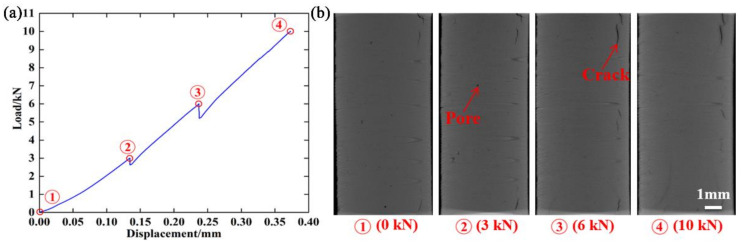
(**a**) Typical load–displacement curve; (**b**) two-dimensional XCT images at different compressive loads.

**Figure 5 materials-15-02547-f005:**
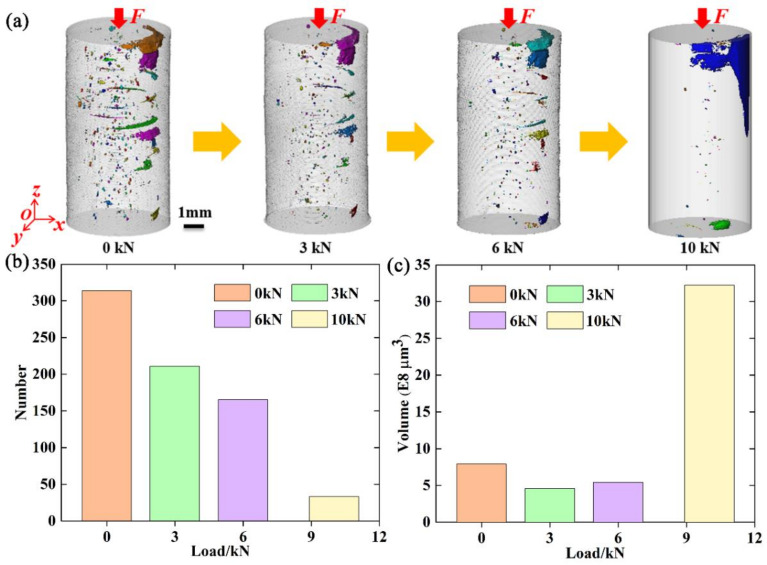
(**a**–**c**): Three-dimensional volume rendering, number, and volume, respectively, of total flaws at different compressive loads.

**Figure 6 materials-15-02547-f006:**
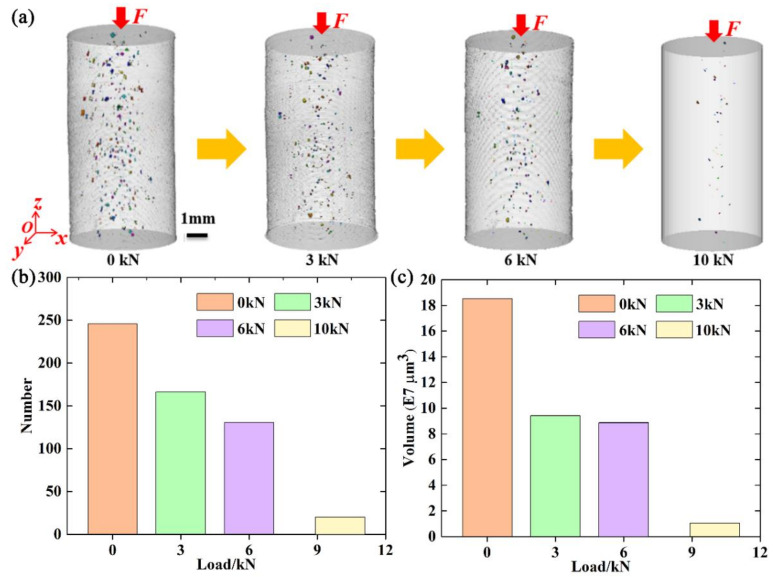
(**a**–**c**): Three-dimensional volume rendering, number, and volume, respectively, of the pores at different compressive loads.

**Figure 7 materials-15-02547-f007:**
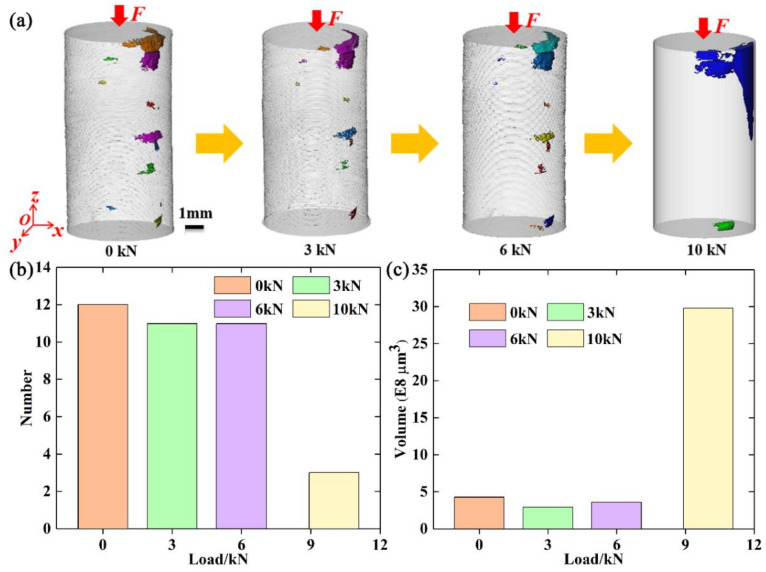
(**a**–**c**): Three-dimensional volume rendering, number, and volume, respectively, of the vertical cracks at different compressive loads.

**Figure 8 materials-15-02547-f008:**
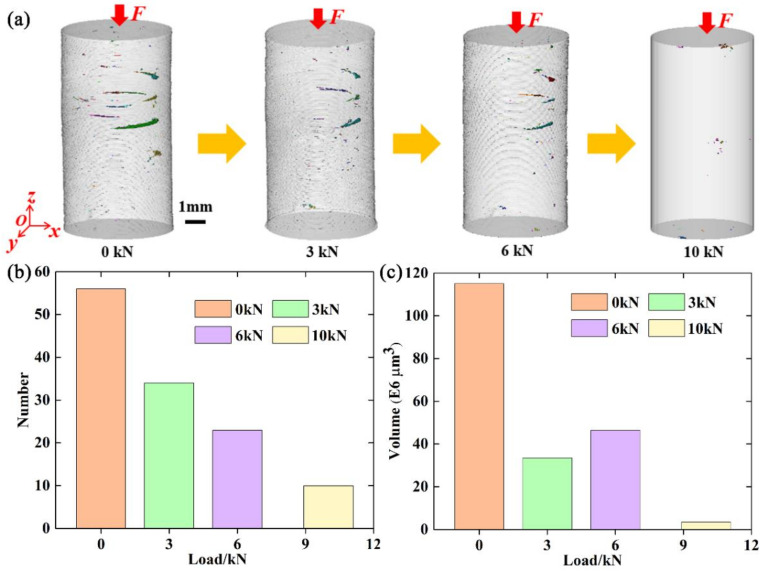
(**a**–**c**): Three-dimensional volume rendering, number, and volume, respectively, of the transverse cracks at different compressive loads.

**Figure 9 materials-15-02547-f009:**
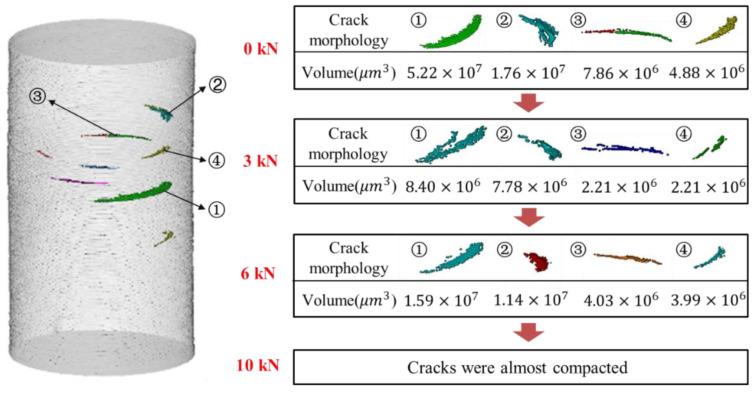
(**a**) Schematic of representative transverse cracks; (**b**) volumes of representative transverse cracks.

**Figure 10 materials-15-02547-f010:**
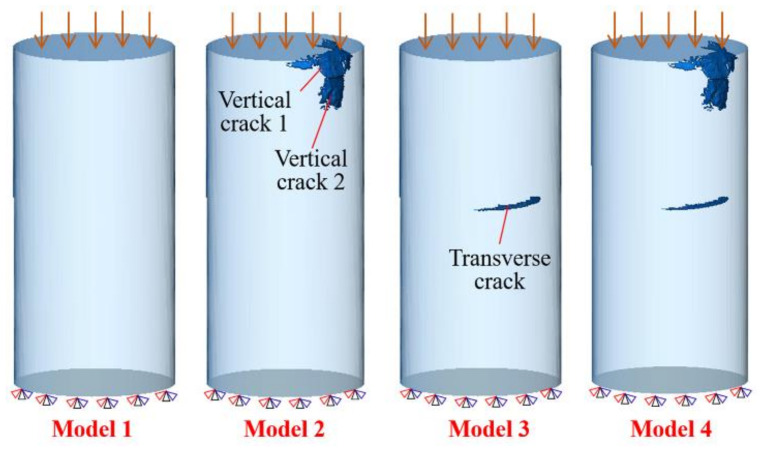
Three-dimensional finite element models.

**Figure 11 materials-15-02547-f011:**
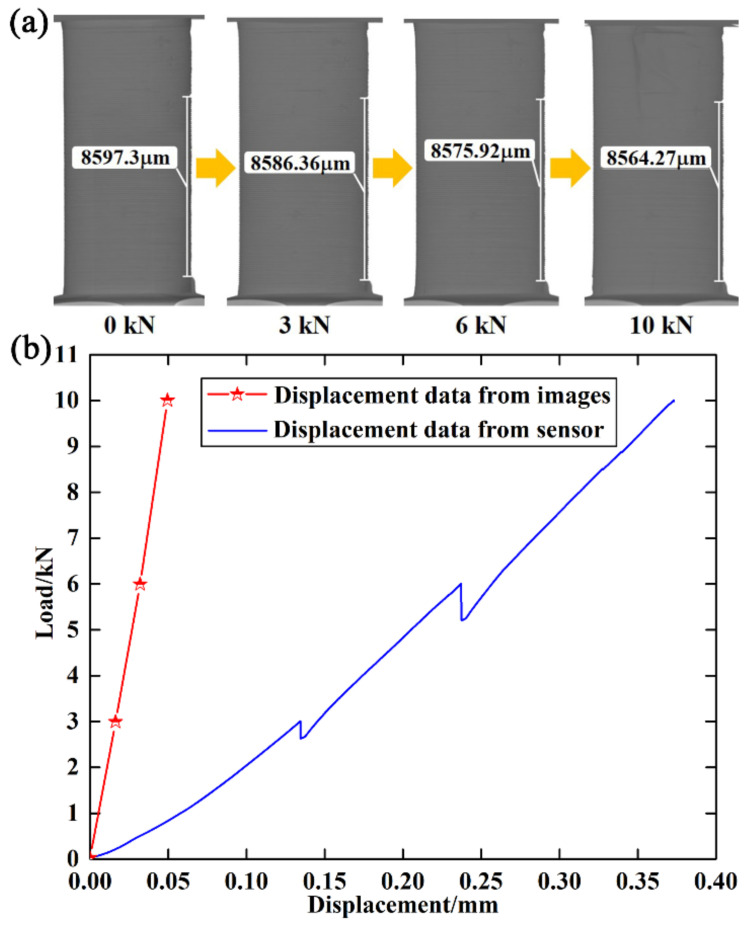
(**a**) Calculation of true displacement; (**b**) load–displacement curves based on the sensor and image data.

**Figure 12 materials-15-02547-f012:**
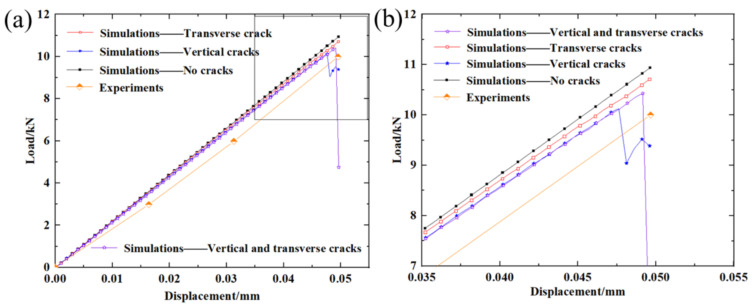
Load–displacement curves obtained from finite element analysis and experiments.(**a**) The whole loading process; (**b**) The section approaching destruction.

**Figure 13 materials-15-02547-f013:**
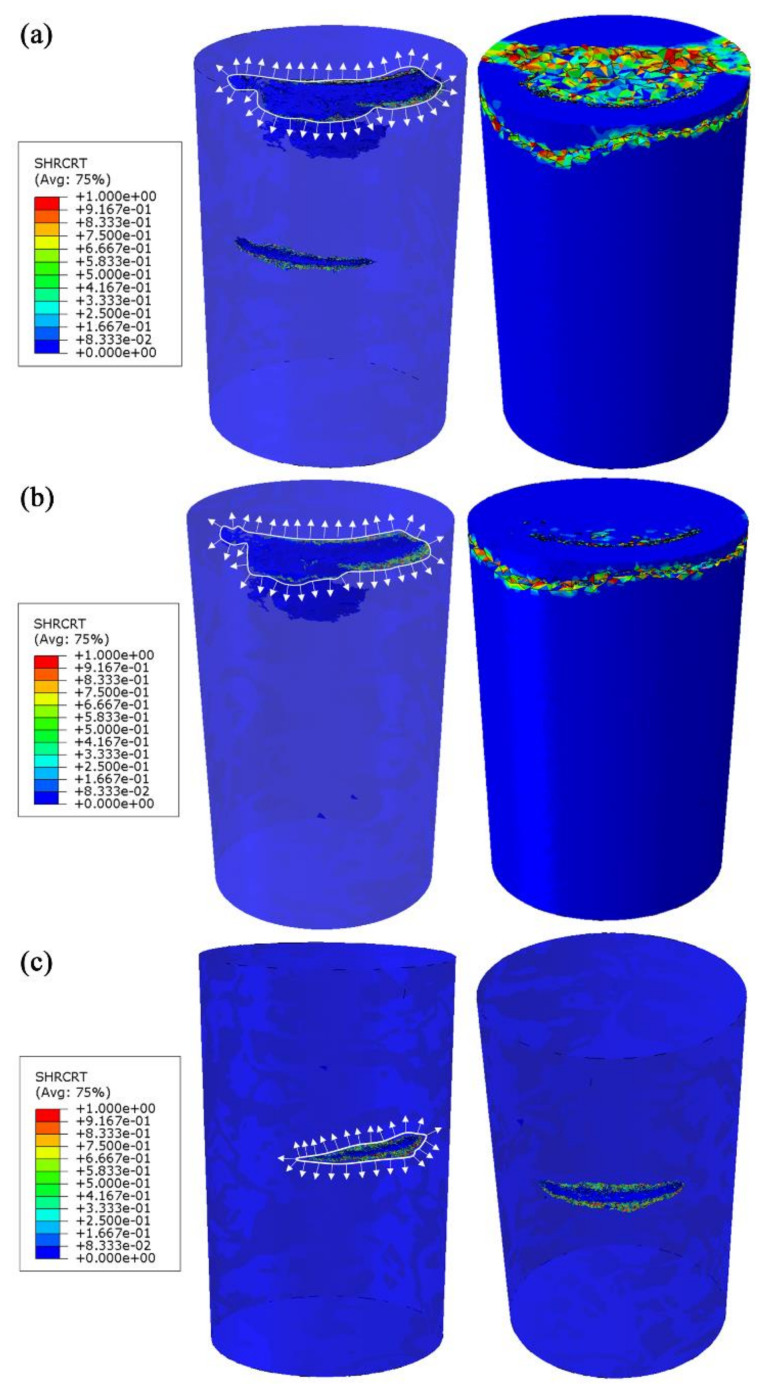
Damage variable distributions for Model 4 (**a**), Model 2 (**b**), and Model 3 (**c**).

## Data Availability

All the supporting and actual data are presented in the manuscript.

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
