# Peer review of "Investigation on Flaw Evolution of Additively Manufactured Al2O3 Ceramic by In Situ X-ray Computed Tomography"

_materials, 2022, doi:10.3390/ma15072547_

Round 1

Reviewer 1 Report

The article Investigation on flaw evolution of additively manufactured Al2O3 ceramic by in-situ X-ray computed tomography is devoted to the study of structural defects in aluminum ceramics using X-ray tomography. Undoubtedly, the results presented by the authors are of high scientific novelty and practical significance, and are also promising for practical research. In general, the presented results of the study can be accepted for publication after the authors provide answers to all the questions raised by the reviewer during the reading of the article.

1. In the abstract, the authors need to more clearly state the purpose and relevance of this work.
2. The authors should present the results of measuring the grain sizes in the samples under study.
3. The authors should explain what caused the choice of such large values ​​of mechanical loads, and also explain where such loads can occur in the future.
4. The authors should explain what is the reason for the formation of a defective structure in the form of pores and microcracks near the surface, according to the presented model.
5. Conclusion requires significant revision.

Author Response

Detailed Responses to Reviewer 1 Dear editor and reviewers, Thank you for your useful comments and suggestions on our manuscript. We have revised the manuscript accordingly, and the detailed responses and corrections are listed below point by point: Reviewer #1: The article “Investigation on flaw evolution of additively manufactured Al2O3 ceramic by in-situ X-ray computed tomography” is devoted to the study of structural defects in aluminum ceramics using X-ray tomography. Undoubtedly, the results presented by the authors are of high scientific novelty and practical significance, and are also promising for practical research. In general, the presented results of the study can be accepted for publication after the authors provide answers to all the questions raised by the reviewer during the reading of the article. Comment 1 In the abstract, the authors need to more clearly state the purpose and relevance of this work. √ Response: Thank you for your useful suggestion. The purpose and relevance of this work has been highlighted in the abstract. The purpose is expressed as “In order to reveal the influence of flaws on mechanical behavior of additively manufactured ceramic, flaw evolution under mechanical loads was studied by in-situ X-ray computed tomography (XCT) in this paper”. The relevance is expressed as “This study may be helpful for process optimization and performance enhancement of additive manufacturing ceramic materials”. The above edits have been highlighted in red in the revised manuscript. Thank you very much for your kind comment. Comment 2 The authors should present the results of measuring the grain sizes in the samples under study. √ Response: Thank you for your useful suggestion. The raw powder contains fine Al2O3 powder (d50=1.05μm) and coarse Al2O3 powder (d50=10.34μm). So, the grain size is large, as shown in Figure 2, the grain size in the sample ranges from 1 to 6 μm. The above edits have been highlighted in red in the revised manuscript. Thank you for your kind comment. Comment 3 The authors should explain what caused the choice of such large values of mechanical loads, and also explain where such loads can occur in the future. √ Response: Thank you for your useful suggestion. To determine compressive loads corresponding to scans, compression tests on five specimens ( 5 mm x 12.5 mm) were conducted. Specimens all failed ranging from 10000 to 15000 N. In order to observe the complete destruction of the specimen, the selected compressive loads included 3000 N, 6000 N and 10000 N. Such loads may occur in large load service environment for structural ceramic components, such as bearings, blade and so on. Thank you for your kind comment. Comment 4 The authors should explain what is the reason for the formation of a defective structure in the form of pores and microcracks near the surface, according to the presented model. √ Response: Thank you for your useful suggestion. The reason is that stress concentration occurs when the cured layer is released from the membrane. Besides, over-cured may also cause high stress due to the high input energy. As the material near the surface is closer to the free end, defective structures in the form of pores and microcracks emerge owing to the stress release during debinding and sintering. The above edits have been highlighted in red in the revised manuscript. Thank you for your kind comment. Comment 5 Conclusion requires significant revision. √ Response: Thank you for your useful suggestion. Conclusion has been rewritten, as follow: This paper investigated the influence of manufacturing flaws on the macroscopic mechanical behavior of additively manufactured Al2O3 ceramic. It is concluded that. (1) The stereolithography additive manufacturing technology was used to prepare Al2O3 ceramic specimens. In-situ XCT compression tests were conducted on additively manufactured Al2O3 ceramic specimen. Three-dimensional full-field morphology of the specimen at 0, 3000, 6000 and 10000 N were obtained by stacking two-dimensional XCT images. (2) Differrnt types of flaws including pores, transverse cracks and vertical cracks, were studied, respectively. The number and volume of the pores, transverse cracks and vertical cracks were analyzed. It is found the number of the flaws tends to decrease with increasing compressive loads, resulting from the compaction effect. At 10000 N, vertical cracks near the upper surface expanded and formed large vertical cracks, leading to the failure of the specimen. (3) Real flaws with morphological and positional information were introduced into the finite element models to study the influence of different types of flaws on mechanical behavior. The true displacements were obtained by analyzing XCT images. The failure mode obtained from the model including two vertical cracks and one transverse crack is similar with the experimental result. It is found that the influence of vertical cracks on the mechanical behavior is greater than that of transverse cracks. The presence of transverse cracks contributes to the evolution of vertical cracks. The above edits have been highlighted in red in the revised manuscript. Thank you for your kind comment. We are pleased to know that our study is of the reviewer’s interest and thank you very much for your critical comments and thoughtful suggestions. We have revised the WHOLE manuscript carefully and re-submitted the new manuscript to your journal. We believe that the manuscript is now suitable for the paper acceptance. The technical and language improvements are highlighted in red in our revised manuscript. The manuscript has been resubmitted to your journal. We are looking forward to your positive response. Yours sincerely, Rujie He, Zhaoliang Qu

Reviewer 2 Report

This paper presents some unique observations of flaws in additively manufactured alumina under compressive load and contrasts them with finite element calculations.  I have a few questions which I hope the authors would address in their revision.

A very common flaw in AM are delaminations in the green state. Where these observed?  None of the flaws in Figure 6a looks like a typical delamination.

What is particle size of the “fine” and “coarse” alumina, and what is the vendor?

Please provide the heating rate for the debinding schedule.

Dimensions of specimen 5 mm diameter 12.5 mm height… was it built with layers in the axial direction or with layers in the transverse direction?  Are these the dimensions of the green specimen or the sintered specimen?

What was the sintered density?  From Figure 2, grain size seems to be about 5 microns.

XCT imaging is done with a voxel size of 8 microns, means that only flaws larger than this can be imaged. Some cracks might be smaller than this—would they be visible?  The smaller flaws in Figure 5a appear to be about 50-100 microns… or comparable to the layer thickness and many times larger than the grain size.  Please discuss whether the imaging is missing some features.

The axial compression of a solid cylinder produces radial tension and shear, which are the stresses that lead to failure by axial splitting or shear fracture.  Is this apparent in the images?

What caused load drops in Figure 3 a.  Apparently this is some sort of relaxation that occurs during the 50 minute XCT scan.  Is this relaxation in the load train or in the specimen?  Also, in view of Figure 10b, which shows that the actual displacement in the specimen is much less than from the displacement sensor, perhaps you should note this to aid the reader.

Figure 6a shows that the XCT field of view was 10 mm high.  Is this the entire height of the sintered specimen?  Is the large flaw at the top of Figure 6a a surface flaw?

There are some things I do not understand about the finite element analysis.  For model 1, with no flaws, are the top and bottom surfaces constrained against radial Poisson expansion?

Author Response

Detailed Responses to Reviewer 2

Thank you for your useful comments and suggestions on our manuscript.

We have revised the manuscript accordingly, and the detailed responses and corrections are listed below point by point:

Reviewer #2:

This paper presents some unique observations of flaws in additively manufactured alumina under compressive load and contrasts them with finite element calculations. I have a few questions which I hope the authors would address in their revision.

Comment 1

A very common flaw in AM are delaminations in the green state. Where these observed? None of the flaws in Figure 6a looks like a typical delamination.

 Response:

The suggestion is useful and helpful. As the specimens were built with layers in the height direction (along z-direction), some transverse cracks can be identified as delaminations. Typical delaminations can be observed in Figure 7. The above edits have also been highlighted in the revised manuscript. Thank you very much for your kind comment.

Figure 7. (a), (b) and (c) Three-dimensional volume rendering, number and volume of the trans-verse cracks at different compressive loads, respectively.

Comment 2

What is particle size of the “fine” and “coarse” alumina, and what is the vendor?

 Response:

We are very sorry for our carelessness. Al2O3 coarse grain particles (3.8 g/cm3, d50 = 10.34 μm, Zhongzhou Alloy Material Co., Ltd., China) were used as the raw material to prepare Al2O3 ceramic slurry. Al2O3 fine grain particles (3.8 g/cm3, d50 = 1.05 μm, Zhongzhou Alloy Material Co., Ltd., China) were added as fine grains. The above edits have been highlighted in red in the revised manuscript. Thank you very much for your kind comment.

Comment 3

Please provide the heating rate for the debinding schedule.

 Response:

We are very sorry for our carelessness. The green bodies were debinded at 550 °C for 2 h (a heating rate of 0.5 °C/min) and pre-sintered at 1000 oC for 2h in a furnace. The heating rate for the debinding schedule was set as 0.5 °C/min. The above edits have been highlighted in red in the revised manuscript. Thank you very much for your kind comment.

Comment 4

Dimensions of specimen 5 mm diameter 12.5 mm height… was it built with layers in the axial direction or with layers in the transverse direction?  Are these the dimensions of the green specimen or the sintered specimen?

 Response:

We are very sorry for our carelessness. The specimen was built with layers in the axial direction and these are the dimensions of the sintered specimen. The above edits have been highlighted in red in the revised manuscript. Thank you very much for your kind comment.

Comment 5

What was the sintered density? From Figure 2, grain size seems to be about 5 microns.

 Response:

The suggestion is useful and helpful. The relative density of sintered specimen is 96.63±0.52 %. The raw powder contains fine Al2O3 powder (d50=1.05μm) and coarse Al2O3 powder (d50=10.34μm). The grain size in the specimen ranges from 1 to 6 μm. The above edits have been highlighted in red in the revised manuscript. Thank you very much for your kind comment.

Comment 6

XCT imaging is done with a voxel size of 8 microns, means that only flaws larger than this can be imaged. Some cracks might be smaller than this-would they be visible? The smaller flaws in Figure 5a appear to be about 50-100 microns… or comparable to the layer thickness and many times larger than the grain size. Please discuss whether the imaging is missing some features.

 Response:

The suggestion is useful and helpful. In our study, only flaws larger than the voxel size of 8 microns can be imaged. Some cracks smaller than the voxel size of 8 microns is not visible. Thus, the flaws larger than the grain size, and smaller than, comparable to or larger than the layer thickness (about 90 microns) can be observed. Generally, the larger the defect size, the greater the influence on macro-mechanical behavior. Considering the resolution of our equipment, only the influence of defects larger than 8 µm on the macroscopic mechanical behavior was studied. Thank you very much for your kind comment.

Comment 7

The axial compression of a solid cylinder produces radial tension and shear, which are the stresses that lead to failure by axial splitting or shear fracture. Is this apparent in the images?

 Response:

The suggestion is useful and helpful. Indeed, the axial compression of a solid cylinder produces radial tension and shear stresses, which lead to failure by axial splitting or shear fracture. As shown in Figure 12, it is clearly found that cracks extend outward along the edge of the contour. This result from the tension and shear stresses at the edge of the crack. The above edits have been highlighted in red in the revised manuscript. Thank you very much for your kind comment. 

Comment 8

What caused load drops in Figure 3a. Apparently this is some sort of relaxation that occurs during the 50 minute XCT scan. Is this relaxation in the load train or in the specimen? Also, in view of Figure 10b, which shows that the actual displacement in the specimen is much less than from the displacement sensor, perhaps you should note this to aid the reader.

 Response:

Thank you for your useful suggestion. The load relaxation is common in in-situ XCT tests [1, 2]. The relaxation occurs during the constant displacement (holding) periods applied to the specimens during the scanning process. In the periods, the load applied to the specimen drops to ensure a constant displacement. Thus, this relaxation is mainly in the specimen. During the calculation of the actual displacement, the load relaxation was neglected. The above edits have been highlighted in red in the revised manuscript. Thank you very much for your kind comment.

[1] Forna-Kreutzer, J.; Ell, J.; Barnard, H.; Pirzada, T.; Ritchie, R.; Liu, D. Full-field characterisation of oxide-oxide ceramic-matrix composites using X-ray computed micro-tomography and digital volume correlation under load at high temperatures. Materials & Design, 2021, 208, 109899.

[2] Bale, H.; Haboub, A.; MacDowell, A.; Nasiatka, J.; Parkinson, D.; Cox, B.; Marshall, D.; Ritchie, R.; Real-time quantitative imaging of failure events in materials under load at temperatures above 1600 C. Nature materials, 2013, 12(1): 40-46.

Comment 9

Figure 6a shows that the XCT field of view was 10 mm high. Is this the entire height of the sintered specimen? Is the large flaw at the top of Figure 6a a surface flaw?

 Response:

Thank you for your useful suggestion. We have checked that the XCT field of view was 12.5 mm high. This is the entire height of the sintered specimen. The large flaw at the top of Figure 6a is a surface flaw, which extends from the surface to the inside of the specimen. Thank you very much for your kind comment.

Comment 10

There are some things I do not understand about the finite element analysis.  For model 1, with no flaws, are the top and bottom surfaces constrained against radial Poisson expansion?

 Response:

Thank you for your useful suggestion. For model 1, with no flaws, all the displacement and rotational degrees of freedom for the bottom surface of the specimen were constrained. A series of downward displacements were imposed on the upper surface of the specimen. The rest of the degrees of freedom for the upper surface were all constrained. Thus, the top and bottom surfaces are both constrained against radial Poisson expansion. Thank you very much for your kind comment.

We are pleased to know that our study is of the reviewer’s interest and thank you very much for your critical comments and thoughtful suggestions. We have revised the WHOLE manuscript carefully and re-submitted the new manuscript to your journal. We believe that the manuscript is now suitable for the paper acceptance. 

The technical and language improvements are highlighted in red in our revised manuscript.

The manuscript has been resubmitted to your journal.

We are looking forward to your positive response.

Yours sincerely,

Rujie He, Zhaoliang Qu

Reviewer 3 Report

The paper entitled ‘Investigation on flaw evolution of additively manufactured Al2O3 ceramic by in-situ X-ray computed tomography’ is suitable for the journal of Materials. The manuscript is exhaustively written and the results are supported by well prepared and discussed experimental and FEM data. The authors performed compression tests with in-situ XCT on additively manufactured Al2O3 ceramics. They investigated flaw evolution behaviours under compressive loads and discussed the influence of different types of flaws on mechanical behaviour. Additionally, authors adopted the finite element analysis to further investigate the influence of different types of flaws on mechanical behaviour of additively manufactured Al2O3 ceramic specimen. As the authors underlined the presented study represent the promising base for the further process optimization and performance enhancement of additive manufacturing ceramic materials.

Author Response

Detailed Responses to Reviewer 3

Thank you for your useful comments and suggestions on our manuscript.

We have revised the manuscript accordingly, and the detailed responses and corrections are listed below point by point:

Reviewer #3:

Comment

The paper entitled ‘Investigation on flaw evolution of additively manufactured Al2O3 ceramic by in-situ X-ray computed tomography’ is suitable for the journal of Materials. The manuscript is exhaustively written and the results are supported by well prepared and discussed experimental and FEM data. The authors performed compression tests with in-situ XCT on additively manufactured Al2O3 ceramics. They investigated flaw evolution behaviours under compressive loads and discussed the influence of different types of flaws on mechanical behaviour. Additionally, authors adopted the finite element analysis to further investigate the influence of different types of flaws on mechanical behaviour of additively manufactured Al2O3 ceramic specimen. As the authors underlined the presented study represent the promising base for the further process optimization and performance enhancement of additive manufacturing ceramic materials.

We are pleased to know that our study is of the reviewer’s interest and thank you very much for your critical comments and thoughtful suggestions. We have revised the WHOLE manuscript carefully and re-submitted the new manuscript to your journal. We believe that the manuscript is now suitable for the paper acceptance. 

The technical and language improvements are highlighted in red in our revised manuscript.

The manuscript has been resubmitted to your journal.

We are looking forward to your positive response.

Yours sincerely,

Rujie He, Zhaoliang Qu

Reviewer 4 Report

The present study reports “Investigation on flaw evolution of additively manufactured Al2O3 ceramic by in-situ X-ray computed tomography”. To make this paper publishable the authors need to consider following comments:

-The greater effect of vertical cracks than transverse is inevitable under axial compression; please mention some more results (outcome of paper) in very end sentences of abstract. Also, there is no name of Stereolithography in Abstract or keywords sections.

-In section 2.1., please mention the alumina powder provider. Also, I think it’s good to have a schematic figure for this section (powder SEM, mixing process, stereolithography and sintering, etc.).

-Please check the scale bar in Figure 2 with higher magnification. for me it seems can be in nm scale. Also Figure 3b needs a scale bar.

-The sentence “It is believed that this study…” not belongs to conclusion; please move it to end of introduction.

-It’s a precisely written work and continuing of authors’ previous article (Ref 8) and I think it’s publishable with minor amendments. 

Author Response

Detailed Responses to Reviewer 4

Thank you for your useful comments and suggestions on our manuscript.

We have revised the manuscript accordingly, and the detailed responses and corrections are listed below point by point:

Reviewer #4:

The present study reports “Investigation on flaw evolution of additively manufactured Al2O3 ceramic by in-situ X-ray computed tomography”. To make this paper publishable the authors need to consider following comments:

Comment 1

The greater effect of vertical cracks than transverse is inevitable under axial compression; please mention some more results (outcome of paper) in very end sentences of abstract. Also, there is no name of Stereolithography in Abstract or keywords sections.

 Response:

Thank you for your useful suggestion. The result “The presence of transverse cracks contributes to the evolution of vertical cracks.” has been added in very end sentences of Abstract. The name of Stereolithography has also been added in Abstract or Keywords sections. Thank you very much for your kind comment.

Comment 2

In section 2.1., please mention the alumina powder provider. Also, I think it’s good to have a schematic figure for this section (powder SEM, mixing process, stereolithography and sintering, etc.).

 Response:

We are very sorry for our carelessness. Al2O3 coarse grain particles (3.8 g/cm3, d50=10.34 μm, Zhongzhou Alloy Material Co., Ltd., China) were used as the raw material to prepare Al2O3 ceramic slurry. Al2O3 fine grain particles (3.8 g/cm3, d50 = 1.05 μm, Zhongzhou Alloy Material Co., Ltd., China) were used as fine grains. The schematic for the preparation of additively manufactured Al2O3 ceramics is plotted in Figure 1. The above edits have been highlighted in red in the revised manuscript. Thank you very much for your kind comment.

Figure 1. Schematic for the preparation of additively manufactured Al2O3 ceramics

Comment 3

Please check the scale bar in Figure 2 with higher magnification. For me it seems can be in nm scale. Also Figure 3. (b) needs a scale bar.

 Response:

Thank you for your useful suggestion. We have checked the scale bar in Figure 2 with higher magnification. The original scale bar in SEM has been added in Figure 2. The scale is mm scale. Besides, a scale bar has also been added in Figure 3. (b). Thank you very much for your kind comment.

Comment 4

The sentence “It is believed that this study…” not belongs to conclusion; please move it to end of introduction.

 Response:

The suggestion is useful and helpful. We have moved the sentence “It is believed that this study…” to end of introduction. Thank you very much for your kind comment. 

Comment 5

It’s a precisely written work and continuing of authors’ previous article (Ref 8) and I think it’s publishable with minor amendments.

 Response:

Thank you very much for your kind comment.

We are pleased to know that our study is of the reviewer’s interest and thank you very much for your critical comments and thoughtful suggestions. We have revised the WHOLE manuscript carefully and re-submitted the new manuscript to your journal. We believe that the manuscript is now suitable for the paper acceptance. 

The technical and language improvements are highlighted in red in our revised manuscript.

The manuscript has been resubmitted to your journal.

We are looking forward to your positive response.

Yours sincerely,

Rujie He, Zhaoliang Qu

Round 2

Reviewer 1 Report

The authors answered all the questions, the article can be accepted for publication.

Author Response

We have recruited the assistance from experienced scientists who are professional English editorial services. The language of the whole manuscript has been smoothed and polished thoroughly. A reputable text editing service suggested by the MDPI English Editing has been used. All the language and grammar edits made to the revised manuscript have been highlighted in red. Thank you for your kind comment.
